# Conditional cooperation in group contests

**Hubert János Kiss** [1,2] ⊙ *, **Alfonso Rosa-Garcia** [3] ⊙, **Vita Zhukova** [4] ⊙

**1** Center for Economic and Regional Studies, Institute of Economics (KRTK KTI), Budapest, Hungary,
**2** Department of Economics, Corvinus University of Budapest, Budapest, Hungary, **3** Departamento de
Fundamentos del Análisis Económico, Universidad de Murcia, Murcia, Spain, **4** Department of Business,
Universidad Católica San Antonio de Murcia, Murcia, Spain

⊙ These authors contributed equally to this work.
* kiss.hubert.janos@krtk.mta.hu

KINGDOM

**Data Availability Statement:** The data underlying
this study are available at Open Science
Framework. The data can be accessed via https://
osf.io/kg9xp/, and the calculations via https://osf.io/
2qhbr/.

## Abstract

In this paper we show experimentally that conditional cooperation, a phenomenon described
in the private provision of public goods, is also present in group contests, where participants'
contributions to their group performance partially determines if they overcome a rival group.
This environment allows us to identify new determinants of conditional cooperation. We
observe conditional cooperation in successful groups and in groups where members contribute more than rivals (even if they lose), but it vanishes in those groups that lose the contest due to low group performance. A random-effect linear panel regression analysis with an
extensive set of controls confirms the findings.

## 1 Introduction

Conditional cooperation is the tendency of individuals to engage in cooperation depending on
the degree of cooperation of other individuals, and is argued to be one of the main sources of
high contributions in social dilemmas [1]. Numerous lab experiments [2–7] have documented
the existence of conditional cooperation using the public goods game. In an overview, [8]
reports that the findings are quite stable across studies, about 61% of participants being classified as conditional cooperators (followed by about 20% of free-riders). This behavior has been
described in other related environments, for instance, in collective-risk social dilemmas (a variant of public goods games), where a group must achieve a given threshold through common
contributions to avoid a general loss (i.e., as a climate change environment), see [9–11]. It is
thus natural to assume that this behavior should at least partly explain how individuals behave
in group contests, i.e., situations in which members of a group face a social dilemma when
competing with other rival groups. [12–15] provide a nice introduction to the theory of group
contest.

Group contests are pervasive, including rent-seeking and lobbying, innovation tournaments and R&D races or sports competitions. The experimental group contest literature consistently finds that average effort level (though often showing a declining pattern) is
significantly higher than the equilibrium prediction, a phenomenon known as overexpenditure, see for instance [16–23]. Some explanations provided by the literature are pure joy of
winning [24–27], bounded rationality [28–30], relative payoff maximization [31] and social

**Funding:** This study was funded by the following: HJK - K 119683, the National Research, Development Innovation (NKFIH) HJK - ECO2017-82449-P, Spanish Ministry of Economy, Industry and Competitiveness HJK - 20764-3/2018/FEKUTSRTAT, Higher Education Institutional Excellence Program of the Ministry of Human Capacities in the framework of the 'Financial and Public Services' at Corvinus University of Budapest. ARG -ECO2016-76178-P, Spanish Ministry of Economy, Industry and Competitiveness ARG - PID2019-107192GB-I00 (AEI/10.13039/501100011033), Spanish Ministry of Economy, Industry and Competitiveness The funders had no role in study design, data collection and analysis, decision to publish, or preparation of the manuscript.

**Competing interests:** The authors have declared that no competing interests exist.

identity [19]. Some studies investigate how participants react to feedback information about others' contribution in individual contests. For instance, [32] finds that individuals ranking higher (lower) decrease (increase) their contribution. Similar findings were reported by [31, 33] in a different setup. [7] reports that most of the participants in individual contests with fixed groups behave reciprocally to opponents' previous choices. Surprisingly, the role of conditional cooperation in group contests has not been analyzed yet. In this paper we approach experimentally this question.

The first question when studying conditional cooperation in group contests is if such a behavior is still present under the simultaneous presence of cooperation and competition. Moreover, the group contest environment makes possible to test how conditional cooperation is shaped by the competitive elements. In this environment, with the lottery contest success function in place, the success of a group in the contest is due to a mixture of group behavior and randomness. This allows us to test if conditional cooperation is affected by 1) the group efforts (that may be larger or lower than those of the rival group) and 2) winning or losing the contest just due to randomness.

To this aim, we carried out a laboratory experiment where subjects played a group contest. Individuals where matched in groups of four subjects and each group played a contest against a rival group repeatedly during 20 rounds. Subjects contributed from their individual endowment to generate the group total contribution. Individual contributions were added up linearly (known in the literature as a perfect substitution performance function). The probability of winning the contest was proportional to the share of the group contribution in the sum of the two groups' contribution (known as the lottery contest success function, CSF hereafter). Finally, the prize obtained by the winner group was shared equally among the group members (known as the egalitarian sharing rule). We say that a group won the contest deservedly (by chance) if the winning group's total contribution was larger (lower) than the rival group's total contribution. Similarly, a group lost deservedly (by chance) if the group lost having a lower (larger) group total contribution than the rival group's total contribution. After each round, the subjects received information about a) their group's total contribution, b) the rival group's total contribution, and c) the winner of the contest. This informational setup allows us to isolate how individuals reacted a) to be in the winner or loser group, b) to be in a group that had a larger or a lower total group contribution than the rival group. Hence, it gives us the opportunity to test to which extent conditional cooperation is affected by these conditions.

As an illustration, consider the following example. In a group of 4 individuals, member A contributes 200 tokens in a given round. She knows that the total contribution of her own group was 1500 tokens, while that of the rival group was 1600 tokens. She knows also that the rival group won the contest. In this case, member A is aware that her contribution was less than the average of her group ($200 < 1500/4 = 375$) and that her group lost deservedly as they accumulated less tokens than the rival group. In the next round, when deciding how much to contribute to the group performance from her endowment, member A may be affected by the fact that a) she contributed less than the average contribution in her group; b) they have lost the contest; c) her group accumulated less tokens than the rival group. The first factor is the standard conditional cooperation argument studied in social dilemma games (see, for instance, [3, 4]) that assumes that individuals tend to conform to the others in their group when deciding how much to contribute to the cooperative effort. This may be driven by social preferences like altruism [34], fairness [35] or inequality aversion [36]. The second factor (winning or losing the contest) may affect the participants in several ways. On the one hand, they may derive non-monetary utility from winning the contest [27] or they may be driven by relative payoff maximization [24, 31]. On the other hand, the aforementioned social preferences affecting cooperation may be enhanced through the contest, participants having stronger feelings

toward in-group members and being hostile toward members of the rival group as proposed by theories like parochial altruism [37–39] or social identity [19, 22, 40, 41]. Note that the contest may strengthen or weaken the effect of standard conditional cooperation. For instance, in our example member A may feel bad having contributed less than the average to the group performance and this feeling is exacerbated by the fact that the group lost. Hence, in the next round she may feel urged to increase her contribution to conform to the others in the group and to increase the probability of winning the contest. However, imagine that member A's group wins the contest in the previous example. Then her desire to conform to the others may be mitigated by the fact that her group won in spite of her lower-than-average contribution. Thus, it is an empirical question to find out how the competitive element affects conditional contribution. The third factor (winning or losing deservedly or not) adds a nuance to the effect of winning or losing. If a group wins after having contributed more than the rival group, then a group member may feel more comfortable than if the group wins by chance. In the latter case, a participant may be more inclined to change her behavior in order to increase the probability of winning in the next round. The opposite argument applies to the case of losing, because losing after having accumulated more tokens than the rival group (that is, by chance) may feel better than losing deservedly (that, in turn, may urge participants to change their contributions in the next round).

Based of the former arguments, we formulate the following conjectures. If a participant contributed less (more) than the average contribution in her group in the previous round, then in the current round she will increase (decrease) her contribution as an attempt to move toward the group average. If the participant contributed less than the group average in the previous round, then her behavior to conform with the group average is strengthened if her group lost the contest. Having lost deservedly (by chance) may make her more (less) likely to increase her contribution. If her group won the contest, then she may be less willing to conform to the group average. This lack of willingness is stronger (weaker) if her group won deservedly (by chance). We expect the opposite if the participant contributed more than the group average in the previous round. She will decrease her contribution, and more so if her group won. Having won deservedly (by chance) makes her decrease her contribution more (less). If her group lost in the previous round, then her desire to decrease her contribution may be mitigated. Having lost deservedly (by chance) makes her decrease her contribution less (more).

We summarize the conjectures in Table 1. Note that we expect that the participants are most likely to increase their contributions if they contributed less than the group average in the previous round and their group lost deservedly, because in this case all the factors that we expect to affect contribution point toward increasing the contribution. Similarly, we expect

**Table 1. Conjectures: Reaction to group average in the previous round, conditioned by winning / losing, deservedly / by chance.**

|  | Effect 1 |  | Effect 2 |  | Effect 3 |
|---|---|---|---|---|---|
| Contribution < group average in *t-1* | *positive* | Her group won in *t-1* | *negative* | deservedly | *negative* |
|  |  |  |  | by chance | *positive* |
|  |  | Her group lost in *t-1* | *positive* | deservedly | *positive* |
|  |  |  |  | by chance | *negative* |
| Contribution > group average in *t-1* | *negative* | Her group won in *t-1* | *negative* | deservedly | *negative* |
|  |  |  |  | by chance | *positive* |
|  |  | Her group lost in *t-1* | *positive* | deservedly | *positive* |
|  |  |  |  | by chance | *negative* |

Positive: increase contribution, negative: decrease contribution.

that participants are most likely to decrease their contributions if they contributed more than the average in the previous round and they won deservedly. We expect the least reaction to having contributed less (more) than the group average when the group won (lost) deservedly.

In the following section we describe our experiment, then we present the results and finally we conclude.

## 2 The experiment

We ran a session at the LINEEX lab in Valencia in July, 2018. The share of males was 41.1%, and participants had a diverse background. The composition of the subject pool according to field of study was the following: 29% Social Sciences and Law, 27% Health Sciences, 20% Engineering and Architecture, 13% Business and / or Economics, 7% Arts and Humanities, 4% Science. The session started with the group contest, followed by experimental games to gather information about the participants' characteristics and a questionnaire. More concretely, we measured social attitudes using the social value orientation task [42], cooperativeness using the public goods game, risk preferences using the bomb risk elicitation task [43], competitiveness using the competitiveness game á la Niederle-Vesterlund [44]. There was no feedback on performance between these experimental measurements. Complete instructions are in S4 Appendix.

Participants knew that they would be paid for the group contest and from the other experimental games the computer would pick randomly one to be paid. At the beginning of the experiment the participants received a consent sheet that they read and signed (if they agreed) before starting the experiment. This written consent contained information about the experiment, the confidentiality of the data and the anonymity of the decisions. No minors were involved in the experiment.

For the group contest, 14 groups of four were formed randomly and anonymously. Although making uncertain the number of rounds would have avoided the last-round effect, in order to have a setup comparable with previous experiments on group contests, we followed [16], so groups were fixed for the 20 rounds of the group contest and the rival group remained the same as well. Participants were endowed with 1000 tokens at the beginning of each round. They could buy competition tokens for their groups, one competition token costing one token. Unused tokens added to the payoff of the participant. We used the lottery CSF, so in each round the probability of winning the contest was proportional to the total competition tokens of a given group divided by the competition tokens of both groups. The winner group received a prize of 4000 tokens, each member obtaining an equal share (1000 tokens). At the end of each round, each participant received information i) on the amount of competition tokens that she bought; ii) on the total amount of competition tokens of the group; iii) on the total amount of competition tokens of the rival group; iv) on whether the group the participant belonged to had won the contest; v) on the individual payoff in the round. Note, therefore, that the first round was informatively different from the rest of rounds, because subjects did not have the additional information on the results of the previous round.

Earnings in the group contest amounted to the sum of the payoffs of 5 randomly chosen rounds (as, for instance, in [19, 28]). Overall, the experiment lasted two hours and participants earned 16 euros on average (including the show-up fee of 5 Euro and the payment for the experimental games to elicit participants' characteristics).

## 3 Findings

Fig 1 indicates the share of participants who decided to increase or decrease their contribution depending on their behavior with respect to the group in the previous round, averaged over all

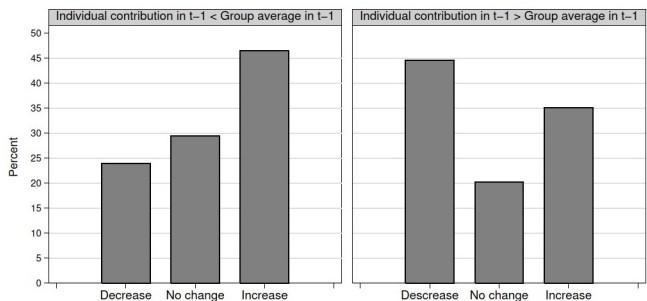

**Fig 1. Share of participants decreasing / not changing / increasing their contribution depending on if their contribution was less (N = 593) / more (N = 464) than the group average in the previous round.** Averaged over all data points.

decisions. (In Fig 1 we consider all the decisions unconditionally, while Table 2 also indicates the corresponding numbers conditional on winning / losing and the nature of winning and losing. Hence, Fig 1 depicts the row *overall* of the top and bottom panels in Table 2). Fig 1 shows that participants increase their contribution more frequently after having contributed less than the group average in the previous round, and that they decrease their contribution more frequently after having contributed more than the group average in the previous round. This suggests that there is conditional contribution in the group contest, the phenomenon being more emphatic upon having contributed less than the group average in the previous round.

In Table 2 we show how participants reacted to when they contributed less / equal / more than the group average in the previous round (that we denote as round *t-1*), conditional on the result of the group contest. Note that subjects knew their own contribution and the group's total contribution, so they could infer if they contributed more or less than the group average. We conjecture that in a group contest the result of the competition may affect the reactions (see Table 1). That is, if a group won or lost the contest in the previous round, then it may influence individual contribution (see the lines *winner / loser*). Since participants knew their own and the rival group's total contribution, they also knew if winning / losing was due to either gathering more competition tokens than the rival group (we call that deserved winner / chance loser) or having made less contribution than the rival group (chance winner / deserved

**Table 2. Participants' reaction to if their contribution was less / more than the group average in the previous round.**

| | Decrease contribution | No change in *t* | Increase contribution | # observations |
|---|---|---|---|---|
| *Contribution < Average in t-1* | | | | |
| overall | 23.95% | 29.51% | 46.54% | 593 |
| winner / loser | 20.14% / 27.54% | 36.46% / 22.95% | 43.40% / 49.51% | 288 / 305 |
| deserved winner / loser | 21.66% / 23.30% | 36.31% / 22.73% | 42.04% / 53.98% | 157 / 176 |
| chance winner / loser | 18.32% / 33.33% | 36.64% / 23.26% | 45.04% / 43.41% | 131 / 129 |
| *Contribution = Average in t-1* | | | | |
| overall | 42.86% | 28.57% | 28.57% | 7 |
| *Contribution > Average in t-1* | | | | |
| overall | 44.61% | 20.26% | 35.13% | 464 |
| winner / loser | 42.26% / 47.11% | 25.52% / 14.67% | 32.22% / 38.22% | 239 / 225 |
| deserved winner / loser | 42.45% / 34.96% | 29.5% / 8.13% | 28.06% / 56.91% | 139 / 123 |
| chance winner / loser | 42% / 61.76% | 20% / 22.55% | 38% / 15.69% | 100 / 102 |

loser). Table 2 depicts all these cases, where we provide the percentage of subjects that choose to increase or decrease their contribution in round *t* given their behavior with respect to their group in *t-1*.

Those who contributed the average amount, tend to decrease their contribution. The most frequent choice of participants who contributed less (more) than the group average in the previous round is to increase (decrease) their contributions, in line with the idea of conditional cooperation, though the relative frequency generally is below 50%. This finding holds for participants in loser and winner groups as well, the effect being somewhat stronger when contribution was less than the group average in *t-1*. Belonging to a winner or loser group by chance or deservedly also affects how participants react to having contributed more or less than the group average in round *t-1*. As expected, the share of those who increased their contribution was highest among those who contributed less than the group average in the previous round and who were in a group that lost deservedly. In fact, in this case the relative frequency of those who increased contribution rises above 50%. However, even in those cases when somebody contributed less than the group average in the previous round (we expect that they would increase their contribution, see Table 1), even if she was in a group that won (which has a negative effect on contribution, according to Table 1) deservedly (again having a negative effect on contribution, see Table 1), increasing the contribution has the highest relative frequency, but in this case well below 50%.

If somebody contributed more than the average, then we expect that she would decrease her contribution according to conditional cooperation. As already commented, overall and also when considering being in a winner and loser group the relatively most frequent reaction is in line with this conjecture. We expected to see the strongest effect when a group wins deservedly (see Table 1), however, contrary to our conjecture, the relative frequency in that case is not the highest. We observe the highest share decreasing their contribution in case of those who were in groups that lost by chance. The only case contradicting the idea of conditional cooperation occurs for those who contributed more than the group average in round *t-1* and were in a group that lost after accumulating less competition tokens than the rival group (deserved loser). In more than 50% of this case contributions increased, even if participants contributed already more than the group average in the previous round.

Chance losers tend to move toward the group average, independently if they contributed more or less than the group average, the effect being stronger in the latter case. Chance winners' reaction is in line with conditional cooperation, as the most frequent reaction is to move towards the group average.

As a robustness check, we study how conclusions change if we consider contributing more / less than the average if it is at least +/- 20% than the average. S1 Table in S1 Appendix reveals that qualitatively we have the same findings.

The idea of conditional contribution is directional in the sense that conditional contributors move toward the average contribution. For instance, both [4] and [8] define conditional contributors based on if an individual's contribution increases (at least in a weakly monotonic way) in the other individuals' contribution or using correlation measures between own and others' contribution. However, it is also natural to ask how is the size of the moves toward the average. Hence, after seeing the direction of the changes, we turn now to the magnitudes. Table 3 provides information on the average size of the change in contribution joint with the standard deviation in each of the cases. When considering the case of having contributed less than the group average in round *t-1*, we see that size of change (in absolute value) is markedly higher for those who decreased their contribution. This is true overall and also if we condition on winning or losing and on doing so deservedly or by chance. Hence, while upon falling short of the group average in round *t-1* the share of those who increase their contribution in

**Table 3. Mean size of participants' reaction to if their contribution was less / more than the group average in the previous round (Standard deviation in parentheses).**

| | Decrease contribution | Increase contribution | # observations |
|---|---|---|---|
| *Contribution < Average in t-1* | | | |
| overall | -46.33 | 20.22 | 593 |
| | (27.37) | (21.59) | |
| winner / loser | -40.20 /-50.57 | 17.84/22.20 | 288 / 305 |
| | (27.83)/(26.39) | (21.21)/(21.78) | |
| deserved winner / loser | -40.47/-54.31 | 18.34/20.74 | 157 / 176 |
| | (29.32)/(26.10) | (21.84)/(18.65) | |
| chance winner / loser | -39.80/-47.01 | 17.29/24.66 | 131 / 129 |
| | (26.18)/(26.48) | (20.66)/(26.26) | |
| *Contribution = Average in t-1* | | | |
| overall | -81.67 | 7.74 | 7 |
| | (7.64) | (0.84) | |
| *Contribution > Average in t-1* | | | |
| overall | -46.92 | 38.80 | 464 |
| | (30.30) | (32.06) | |
| winner / loser | -42.22/-51.39 | 34.38/42.76 | 239 / 225 |
| | (29.85)/(30.18) | (29.41)/(33.94) | |
| deserved winner / loser | -42.04/ -51.47 | 36.55/40.39 | 139 / 123 |
| | (31.64)/(30.77) | (29.82)/(32.24) | |
| chance winner / loser | -42.48/-51.34 | 32.15/53.17 | 100/ 102 |
| | (27.51)/(30.03) | (29.21)/(40.06) | |

round *t* is higher (and often by a wide margin) than the share of those who decrease, those in the latter group decrease much more their contribution on average than those who increase their contribution. This suggests that even if increasing contribution (the choice in line with the idea of conditional cooperation) is the most frequently observed reaction among those who contributed less than average in round *t-1*, their impact on overall contribution may be neutralized by participants who decrease their contribution in the same condition due to the fact that in the latter group the size of the change is larger.

Turning to those who in round *t-1* contributed more than the average, we observe that in this group those who decrease their contribution (in line with conditional cooperation), do so by a larger extent than those who increase their contribution. This happens in all cases, except for chance losers. Hence, for those contributing more than the average in the previous round, it is not just the share of those who decrease their contribution in the next round is higher, but also the size of their reaction is larger than the size of those who react in the opposite way.

The fact that the magnitude of decreases is larger than the magnitude of increases independently if the contribution in the previous round was less / equal / more than the group average is partly due to the fact that there is a downward trend in the level of contributions (see S2 Appendix) that implies that overall decreases are larger than increases. Interestingly, while the size of the decreases is similar when contribution in the previous round was less or more than the group average, for increases we observe an asymmetry. The magnitude of increases is clearly larger when contribution in the previous round was already above the group average. We have no explanation why this happens. More research is needed to understand these surprising findings.

In order to see if conditional contribution plays a significant role in the participants' decisions, we carry out a random-effect linear panel regression (see Table 4), where we exploit the

**Table 4. Determinants of the percentage change in individual's contribution in *t* with respect to *t-1*.** Random effects linear panel regression model.

| Dependent variable: Percentage change in individual contribution in round *t* with respect to *t-1* | | | | |
|---|---|---|---|---|
| | Spec. 1 | Spec. 2 | Spec. 3 | Spec. 4 |
| i's contribution (*t-1*) w.r.t avg group contribution (*t-1*) in pc | -0.081*** | -0.074*** | 0.001 | -0.060 |
| | (0.021) | (0.021) | (0.037) | (0.039) |
| Round | -0.609*** | -0.628*** | -0.600*** | -0.622*** |
| | (0.222) | (0.221) | (0.221) | (0.219) |
| Winner by chance (*t-1*) | | -2.993 | -3.310 | -2.780 |
| | | (3.455) | (3.448) | (3.428) |
| Winner deserved (*t-1*) | | -4.074 | -5.301 | -5.388 |
| | | (3.243) | (3.274) | (3.353) |
| Loser by chance (*t-1*) | | -13.375*** | -14.593*** | -14.447*** |
| | | (3.471) | (3.498) | (3.554) |
| i's wrt gr's contribution (*t-1*) x winner chance (*t-1*) | | | -0.061 | -0.056 |
| | | | (0.039) | (0.039) |
| i's wrt gr's contribution (*t-1*) x winner deserv (*t-1*) | | | -0.057 | -0.044 |
| | | | (0.044) | (0.044) |
| i's wrt gr's contribution (*t-1*) x loser chance (*t-1*) | | | -0.136*** | -0.109** |
| | | | (0.046) | (0.047) |
| Covariates | | | | YES |
| Constant | 1.147 | 6.187* | 7.674** | -67.432 |
| | (2.742) | (3.391) | (3.437) | (45.978) |
| $R^2$ | 0.020 | 0.034 | 0.043 | 0.079 |
| Observations | 1064 | 1064 | 1064 | 1064 |

Standard errors in parentheses.

Random effects linear panel regression model.

Dependent variable normalized.

* $p < 0.10$,

** $p < 0.05$,

*** $p < 0.01$

panel dimension of our dataset. (Confidence intervals are provided in S2 Table, see S3 Appendix).

The dependent variable is the percentage change in individual contribution in round *t* with respect to *t-1*. Percentage change is computed as the percentage of variation with respect to the maximum possible variation. Since contribution is in the range of [0, 1000], it implies, for instance, that after having contributed 800, changing to 400 is a -50% of variation and changing to 900 is a +50% of variation. We use percentage change because there is a downward trend in contribution in our data (see S1 Fig of S2 Appendix), so a decrease of 100 tokens is not the same in relative terms at the beginning of the experiment as at the end.

The explanatory variable of main interest is the individual contribution in round *t-1* relative to the average group contribution in *t-1* in percentage (measuring percentage as in the dependent variable), so here we are not only interested if an individual contributed more or less than the group average in *t-1*, but we also take into account the degree of deviation from the group average. In all specifications, we also control for round as the change in individual contribution may vary over time. In specification 2 we add the effect of losing or winning deservedly or by chance in the previous round, deserved loser being the baseline case. Note that subjects received information on whether their group had accumulated more competition tokens than

the rival group and if they had won or lost, and therefore information on winning and losing and having done so deservedly or by chance was correlated. However, that information uniquely determined if the group had been a winner or a loser by chance or deservedly, and therefore the included dummy variables are orthogonal by construction. In specification 3 we also add interaction terms to see if the percentage change in individual contribution from *t-1* to *t* is different based on winning or losing the contest after having accumulated more or less competition tokens than the rival group has a differential effect. In the last specification we also use controls related to socio-demographics (female, age, academic degree, number of siblings, body mass index, digit ratio, breadwinner's employment and participant's work per week), IQ variables (being reflective or irreflective in the Cognitive Reflection test [45–47]) and the economic preferences that we elicited in the questionnaire.

In specifications 1 and 2 the coefficient of the main explanatory variable is negative and significant, indicating that having contributed more (or less) than the group average in round *t-1* provokes a move in the opposite direction. We view it as strong evidence on conditional cooperation in contests. Round has always a negative and significant coefficient, showing a downward trend. In specification 2, we find that being in a group that loses by chance is the only group outcome that has a significant effect. The estimated coefficient indicates that it reduces contribution change in 13 pp, relative to deserved loser groups, as conjectured (see Table 1). Contribution change after losing deservedly or winning (by chance or deservedly) is not statistically different, and higher than in the case of being in a group that loses by chance. In specification 3 we interact the group outcomes with the difference between own contribution and the average of the group in round *t-1*. Note that contributing differently than the group average in round *t-1*, the explanatory variable of main interest is not significant any more, however the interaction between this variable and loser by chance is negative and is the only significant new variable. It indicates that the effect of differing from the group average is significantly different when the individual is in a group that loses by chance. These findings hold even if we add the remaining control variables (specification 4).

Specifications 3 and 4 allow us to understand how conditional cooperation depends on the different elements of the environment. In order to check the occurrence of conditional cooperation, we test the combination of the relevant coefficients (see Table 5). We find that conditional contribution matters and participants significantly approximate their contribution to the group's average if they belong to a winner group (either by chance or deservedly) or to a group that loses by chance. However, this effect vanishes (is not significant) in the baseline case, that is, when the group loses deservedly. This is not surprising as we have seen that in

**Table 5. Testing when participants behave as predicted by conditional cooperation.**

| | Spec. 3 | Spec. 4 |
|---|---|---|
| | Coefficient | Coefficient |
| i's contr. (*t-1*) wrt gr + (i's wrt gr X winner by chance) | -0.061* | -0.116*** |
| i's contr. (*t-1*) wrt gr + (i's wrt gr X winner deserved) | -0.056* | -0.104*** |
| i's contr. (*t-1*) wrt gr + (i's wrt gr X loser by chance) | -0.135*** | -0.169*** |
| constant + winner by chance | 4.364 | -70.212 |
| constant + winner deserved | 2.373 | -72.821 |
| constant + loser by chance | -6.920** | -81.879* |

* $p < 0.10$,

** $p < 0.05$,

*** $p < 0.01$

that case participants tend to increase their contributions, independently of how their contributions relate to the average contribution in the previous round (see Table 2).

## 4 Conclusion and discussiom

We report experimental evidence on conditional cooperation being present in group contests as overall the most frequent reaction to having contributed more (less) than the group average in the previous round is to decrease (increase) contribution. We also show that the outcome of the group contest also affects conditional cooperation. Conditional cooperation is not observed in the group contest after losing deservedly, however participants do approximate their behavior to the group's average when being in a winner group or after losing by chance in the previous round. Hence, similarly to public goods game, we document the existence of conditional cooperation in group contest, except when being in a deserved loser group.

While these findings reveal the role of conditonal cooperation in group contests, it is important to consider some limitations. We lack an explanation for some of the asymmetric behaviors observed in the experiment. In particular, the change in contributions is larger when contribution in the previous round was already larger than the group average relative to the case when contribution in the previous round was smaller than the group average. Moreover, though the coefficient of the main variable of interest in Table 4 behaves as expected, the overall explanatory power of the regressions is rather low. This suggests that there are additional important factors that shape decision-making. More research is needed to find out what those factors are and if they are related to conditional cooperation in this environment. Further research should also reveal to what extent the findings depend on specific elements of the environment. Importantly, group contests come in many formats (see [15]) that may affect the behavior of the subjects, and our experiment uses just one of the standard contest setups. For instance, if instead of the lottery CSF the auction CSF is used (when the group with the highest performance wins the contest with certainty) and as a consequence the uncertainty involved in the determination of the winner is eliminated, subjects may be more willing to increase contributions as it is clearer how it augments the chances to win. Hence, we believe that investigating the effect of the elements of group contest on conditional cooperation is a promising venue for future research.

## Supporting information

**S1 Appendix. Supplementary material—Robustness check.**
(PDF)

**S2 Appendix. Supplementary material—Downward trend in contribution.**
(PDF)

**S3 Appendix.**
(PDF)

**S4 Appendix. Supplementary material—Instructions.**
(PDF)

## Author Contributions

**Conceptualization:** Hubert János Kiss, Alfonso Rosa-Garcia, Vita Zhukova.

**Data curation:** Hubert János Kiss, Alfonso Rosa-Garcia, Vita Zhukova.

**Formal analysis:** Hubert János Kiss, Alfonso Rosa-Garcia, Vita Zhukova.

**Funding acquisition:** Hubert János Kiss, Alfonso Rosa-Garcia.

**Investigation:** Hubert János Kiss, Vita Zhukova.

**Methodology:** Hubert János Kiss, Alfonso Rosa-Garcia, Vita Zhukova.

**Software:** Vita Zhukova.

**Writing – original draft:** Hubert János Kiss, Alfonso Rosa-Garcia, Vita Zhukova.

**Writing – review & editing:** Hubert János Kiss, Alfonso Rosa-Garcia, Vita Zhukova.

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
