## [Decision Letter · Decision Letter 0]

18 Sep 2020

PONE-D-20-25168

Conditional cooperation in group contests

PLOS ONE

Dear Dr. Kiss,

Thank you for submitting your manuscript to PLOS ONE. After careful consideration, we feel that it has merit but does not fully meet PLOS ONE’s publication criteria as it currently stands. Therefore, we invite you to submit a revised version of the manuscript that addresses the points raised during the review process.

 The two reviewers have agreed that the paper addresses an interesting problem and has good merit. However, there are still some major issues that need to be addressed. Authors should consider the reviewers' comments and suggestions carefully when preparing the revised version.

We look forward to receiving your revised manuscript.

Kind regards,

The Anh Han, Ph.D.

Academic Editor

PLOS ONE

Journal Requirements:

3. We note you have included a table to which you do not refer in the text of your manuscript. Please ensure that you refer to Table 4 in your text; if accepted, production will need this reference to link the reader to the Table.

Additional Editor Comments:

The two reviewers have agreed that the paper addresses an interesting problem and has good merit. However, there are still some major issues that need to be addressed. Authors should consider the reviewers' comments and suggestions carefully when preparing the revised version.

Reviewers' comments:

Reviewer's Responses to Questions

**Comments to the Author**

1. Is the manuscript technically sound, and do the data support the conclusions?

Reviewer #1: Partly

Reviewer #2: Yes

2. Has the statistical analysis been performed appropriately and rigorously? 

Reviewer #1: Yes

Reviewer #2: Yes

3. Have the authors made all data underlying the findings in their manuscript fully available?

Reviewer #1: No

Reviewer #2: Yes

4. Is the manuscript presented in an intelligible fashion and written in standard English?

Reviewer #1: Yes

Reviewer #2: Yes

5. Review Comments to the Author

Reviewer #1: Overview and general recommendation

This manuscript presents a behavioral economics experimental study that investigates whether conditional cooperation is present in group contests, a type of game in which groups have to cooperate in order to increase their chances of winning a prize when competing against a rival group. The authors argue that their findings evidence the existence of condition cooperation in group contests and that such behavior is correlated with the success of the group.

The research questions pursued in this manuscript are relevant and aim to shed light into the nature of human decision-making in groups, whether this decision-making is conditional and which factors affect such conditional behavior. Overall, the manuscript is well written and addresses an important subject, as well as provide interesting conclusions. However, I have some major concerns related to the experimental design and the results, which prevent me from recommending the acceptance of the article until they are addressed. If the authors are able to address my concerns or convince me otherwise, I would be eager to change my assessment and recommend acceptance.

Major comments

Experimental design

1. In this experiment, if I understand correctly (please correct me if I did not), participants may actually obtain an (almost) egalitarian distribution of payoffs between their own group and the rival groups if they do never contribute: if no one contributes, then each participant will keep their initial endowment and there is a 50% of chances that each group will receive the prize, so on average both groups should receive the prize as many times over the course of the experiment. However, Figure S2 shows that on average participants contributed about 3/10 of their endowment (please note that this Figure lacks confidence intervals and sample size information; this should be added). What do the authors think might be the motivation behind this result? It would be interesting for the manuscript if the authors provided some expected model of behavior for the observed results apart from the hypotheses already indicated.

2. I could not see anywhere what is the participant show-up fee of the experiment, i.e., how much does each participant receive just for showing up. Neither did I see any information about the ethical advice of the experiment. I think the authors should comment on that and also upload the data of this experiment so that reviewers can check the validity of the results.

Experimental results

1. In Table 2, the authors show the percentage of participants that increase/decrease their contributions with respect to some conditions. Yet, the magnitude of the change is not shown, nor whether the results are significant. I encourage the authors to provide this information, so that the statistics presented in the manuscript are more transparent, and it is easier to understand the magnitude of the effect presented in the paper.

The results in this table appear to indicate that there are 22.59% more participants that increase their contributions in comparison to those that decrease them, when the participant had contributed less than the average in the previous round. Likewise, participants that contributed more than the average tend to reduce their contributions in the next round (although the difference here is much smaller, ~9.48%). This seems to confirm the hypothesis that participants tend to approximate their contributions to the average of the group. This is an interesting result by itself. However, the differences between decreasing/increasing contributions between participants who are in groups that won/lost deservedly/by chance are, in general, quite small and it is difficult to see whether this has an effect. Hence, the main effects (the conditional behavior observed for participants that contribute more than the average) should be emphasized.

2. The authors perform a panel regression with different combinations of parameters (specifications) to check significance. They say “In specifications 1 and 2 the coefficient of the first variable is negative and significant, indicating that having contributed more or less than the average and the magnitude of deviation provokes a strong move in the opposite direction”. I agree with the first part of the phrase, but significance does not indicate the magnitude of the effect, thus I do not see how the authors can backup that there is a strong move in the opposite direction (see [1] and [2]). In fact the coefficient of the regression is -0.081.

It would be really helpful to be able to see the R^2 of the regression to understand whether it is actually providing a good fit for the data. Additionally, some of the effects tested are correlated: information about winning/losing the game and about the contributions is shown at the same time to the participants, thus, its effect on their actions in the next round might be correlated. This can hinder the trust in the p-values results.

3. Both in the main text and in the S2 Appendix it appears that the threshold for statistical significance is 0.1. Is that correct? This is not a big issue for me, yet it is common to consider the threshold at 5% and thus the results of the Wilcoxon signed-rank test (p-value=0.085) would not be significant.

Overall, I believe the results presented in the experiment are interesting, yet they need to be stated in a more clear and transparent manner. In particular, Table 2 could benefit from a visual representation for at least some of the results, using, for instance, a bar plot.

Minor comments

The topic of conditional behavior in groups is very interesting. However, the group contest presented in this manuscript has some similarities with threshold public good games, in the sense that it encompasses both cooperative behavior towards the group as well as competition. Particularly, the study presented in [3] shows experimentally how conditional behaviour emerges in this type of games. How does your study compare to it?

There are also some minor typos and format issues in the paper:

1. Line 187: There is a space missing after the point: “[…] to move towards the group average.As a robustness check […]”

2. Line 220: There is something wrong in this phrase: “[…] reduces contribution [change up to 13 pp], relative […]”

3. The font size in the tables should be bigger to increase readability.

4. The table captions should contain more explanations about what is shown. In particular in Table 3, the type of regression performed should be indicated. Authors should also indicate the R^2 value in case this was a linear regression.

References

[1] https://statisticsbyjim.com/regression/interpret-coefficients-p-values-regression/

[2] https://statisticsbyjim.com/regression/low-r-squared-regression/

[1] Domingos, E. F., Grujić, J., Burguillo, J. C., Kirchsteiger, G., Santos, F. C., & Lenaerts, T. (2020). Timing uncertainty in collective risk dilemmas encourages group reciprocation and polarization. arXiv preprint arXiv:2003.07317.

Reviewer #2: In this manuscript, the authors explore the determinants of conditional cooperative behavior in group contests. The results show that conditional cooperative behavior can appear in the successful group and those who contribute more than the rivals. Furthermore, this result is also verified by the random effect linear regression model. This paper has a good motivation.

However, there are some remaining issues with the manuscript, requiring some answers.

1) The setting of the first round of the experiment was unclear.

2) Previous experimental studies have shown that the number of rounds has an effect on the evolutionary results. The authors should discuss why the number of experimental rounds is set as 20 instead of the case that players are not informed the concrete game round.

3) Does t-1 mean previous round? t-1 is used in many places in the article, but I don't understand it. Does it refer to the previous round or all previous rounds? Concretely, see page 5, line 160. In addition, t should be italicized.

4) The probability of players using C and D at the beginning of the public goods game should be explained.

5) There are some improper spaces in the typesetting of manuscripts, which leads to the destruction of the coherence of sentences and affects the reading. In addition, some expressions are not standardized.

6) Using some histogram or line chart can describe the change of contribution level in the game process more clearly.

7) It is better to provide 95% confidence interval when analyzing linear regression model.

6. PLOS authors have the option to publish the peer review history of their article (what does this mean?). If published, this will include your full peer review and any attached files.

Reviewer #1: No

Reviewer #2: No

---

## [Author Response · Author response to Decision Letter 0]

22 Oct 2020

We uploaded the answers to the editor and to the reviewers.

---

## [Decision Letter · Decision Letter 1]

10 Nov 2020

PONE-D-20-25168R1

Conditional cooperation in group contests

PLOS ONE

Dear Dr. Kiss,

Thank you for submitting your manuscript to PLOS ONE. After careful consideration, we feel that it has merit but does not fully meet PLOS ONE’s publication criteria as it currently stands. Therefore, we invite you to submit a revised version of the manuscript that addresses the points raised during the review process.

We look forward to receiving your revised manuscript.

Kind regards,

The Anh Han, Ph.D.

Academic Editor

PLOS ONE

Additional Editor Comments (if provided):

Both reviewers have agreed that revised version has been improved, but their previous comments have only been partially addressed. There are still a few aspects remained to be dealt with as suggested by the reviewers, which the authors should carefully consider in their revision.

Reviewers' comments:

Reviewer's Responses to Questions

**Comments to the Author**

1. If the authors have adequately addressed your comments raised in a previous round of review and you feel that this manuscript is now acceptable for publication, you may indicate that here to bypass the “Comments to the Author” section, enter your conflict of interest statement in the “Confidential to Editor” section, and submit your "Accept" recommendation.

Reviewer #1: All comments have been addressed

Reviewer #2: (No Response)

2. Is the manuscript technically sound, and do the data support the conclusions?

Reviewer #1: Yes

Reviewer #2: Yes

3. Has the statistical analysis been performed appropriately and rigorously? 

Reviewer #1: Yes

Reviewer #2: Yes

4. Have the authors made all data underlying the findings in their manuscript fully available?

Reviewer #1: No

Reviewer #2: Yes

5. Is the manuscript presented in an intelligible fashion and written in standard English?

Reviewer #1: Yes

Reviewer #2: Yes

6. Review Comments to the Author

Reviewer #1: The authors have addressed all my comments. And the new figure added largely help the understanding of the main message of the article. In particular, now, in Figure 1 it is clear that there is a conditional effect with most participants increasing/decreasing their contributions if they previously contributed less / more than the group’s average in the previous round. Thus I recommend that this paper is accepted. Yet, some parts of the paper are still not clear and I would prefer that the authors review these parts before the paper is published:

Figure 1: It is not clear if these are overall results, or if they take into account only the conditions in which players won/lost. This is particularly important since later in Table 2 you do condition the results on win/loss. It would be good to clarify both in the text and the Figure’s Caption that this are results averaged over all data points. Moreover, there should be more information about the sample used in this figure (how many data points are used for each bar – remember you are using percentages).

Table 3: It appears that the magnitude of decrease in all cases is higher than that of increase. Which indicates that in all cases participants are more careful when increasing their contributions. Moreover, the magnitude of increase is bigger when players already contributed more than the average in the previous round, why does this happen? Is it possible that not only the previous round matters in this case?

Also, it seems surprising that the majority of participants that contributed as much as the group average, decrease their contributions with the biggest change in all 3 conditions (>, =, <), and this happens independently on whether the group lost or won the contest. Is there some explanation to this?

Table 4: The R^2 values are quite small. I would like to see at the end of the paper some comment about the limitations of the study, both in this regard and in general. Overall, I do miss this critical overview of the work that help readers understand the limitations of the study and thus apply the conclusions with care.

Figure 2 should be Figure S2. Moreover, it appears that there are no significant individual changes in contributions per round.

Finally, while in the text you name “Figure 1” or “Figure 2”, you label figures as “Fig. 1” and “Fig. 2”, this should be corrected. Moreover, in the text that you have already corrected you write: “In S2 Appendix we present evidence…”. I believe you mean “In Figure S2 of the Appendix, we present evidence…”. It would be good if the authors reread the paper once more to eliminate these typos.

Reviewer #2: The authors have addressed most of my comments, but a few issues with the paper's clarity still remain for me and should be seen to before publication.

In page 1, line 6, "[8] report that..." should be corrected as "[8] reports that...";

In page 2, line26, "[30] find that..." should be corrected as "[30] find that...";

In page 2, line 27, "[7] report that..." should be corrected as "[7] reports that..."

...

The authors should examine the manuscript carefully to avoid these grammatical problems. In addition, there are also some English writing problems in "the experiment" section, especially tenses, which need to be carefully proofread.

In dealing with the problem of promoting the evolution of cooperative behavior, there are many related papers worth reviewing, such as Cost effective external interference for promoting the evolution of cooperation, 8:15997 (2018)，Evolutionary dynamics of cooperation in a population with probabilistic corrupt enforcers and violators, 29(11): 2127­-2149 (2019).

7. PLOS authors have the option to publish the peer review history of their article (what does this mean?). If published, this will include your full peer review and any attached files.

Reviewer #1: No

Reviewer #2: No

---

## [Author Response · Author response to Decision Letter 1]

20 Nov 2020

We attached our responses to the reviewers.

---

## [Decision Letter · Decision Letter 2]

4 Dec 2020

Conditional cooperation in group contests

PONE-D-20-25168R2

Dear Dr. Kiss,

We’re pleased to inform you that your manuscript has been judged scientifically suitable for publication and will be formally accepted for publication once it meets all outstanding technical requirements.

Kind regards,

The Anh Han, Ph.D.

Academic Editor

PLOS ONE

Additional Editor Comments (optional):

Both reviewers are happy with the changes made and have recommended publication. There is only a minor issue that should be addressed when preparing the final version.

Reviewers' comments:

Reviewer's Responses to Questions

**Comments to the Author**

1. If the authors have adequately addressed your comments raised in a previous round of review and you feel that this manuscript is now acceptable for publication, you may indicate that here to bypass the “Comments to the Author” section, enter your conflict of interest statement in the “Confidential to Editor” section, and submit your "Accept" recommendation.

Reviewer #1: All comments have been addressed

Reviewer #2: All comments have been addressed

2. Is the manuscript technically sound, and do the data support the conclusions?

Reviewer #1: Yes

Reviewer #2: Yes

3. Has the statistical analysis been performed appropriately and rigorously? 

Reviewer #1: Yes

Reviewer #2: Yes

4. Have the authors made all data underlying the findings in their manuscript fully available?

Reviewer #1: Yes

Reviewer #2: Yes

5. Is the manuscript presented in an intelligible fashion and written in standard English?

Reviewer #1: Yes

Reviewer #2: Yes

6. Review Comments to the Author

Reviewer #1: The authors have addressed all my previous comments, so I recommend the paper for acceptance.

However, I would like to note that the font size of Tables 1 and 3 is very small, making it barely legible. I recommend that the authors fix that in the publication ready version.

Reviewer #2: Overall, the authors have carefully and thoughtfully addressed all of my concerns, and I think this paper should be published.

7. PLOS authors have the option to publish the peer review history of their article (what does this mean?). If published, this will include your full peer review and any attached files.

Reviewer #1: No

Reviewer #2: No

---

## [Editor Report · Acceptance letter]

9 Dec 2020

PONE-D-20-25168R2 

Conditional cooperation in group contests 

Dear Dr. Kiss:

I'm pleased to inform you that your manuscript has been deemed suitable for publication in PLOS ONE. Congratulations! Your manuscript is now with our production department. 

Kind regards, 

on behalf of

Dr. The Anh Han 

Academic Editor

PLOS ONE